# Attaining the Promise of Geminivirus-Based Vectors in Plant Genome Editing

**DOI:** 10.3390/v17050631

**Published:** 2025-04-27

**Authors:** Muhammad Arslan Mahmood, Muhammad Waseem Sajjad, Ifrah Imran, Rubab Zahra Naqvi, Imran Amin, Muhammad Shafiq, Muhammad Qasim Aslam, Shahid Mansoor

**Affiliations:** 1Research School of Biology, The Australian National University, Canberra, ACT 2601, Australia; 2Department of Biological Sciences, University of Sialkot, Sialkot 51040, Pakistan; 3National Institute for Biotechnology and Genetic Engineering, College of Pakistan Institute of Engineering and Applied Sciences, Faisalabad 38000, Pakistan; 4Department of Biotechnology, University of Management and Technology, Sialkot Campus, Sialkot 51310, Pakistan; 5Department of Biotechnology, Riphah International University Faisalabad Campus, Faisalabad 44000, Pakistan; 6Jamil ur Rehman Center for Genome Research, University of Karachi, Karachi 75270, Pakistan

**Keywords:** geminivirus, biotechnology, genome editing, vector, replicon

## Abstract

Over the last 40 years, several studies have provided evidence demonstrating that viral vectors can result in effective gene targeting/insertions in a host’s genome. The traditional approaches of gene knock-down, -out, or -in involve an intensive transgenesis process that is plagued by extensive timescales. Plant viruses have the potential to target specific genes and integrate exogenous DNA molecules at the target locus. Their ability to manipulate a host’s genetic material and become a part of it makes them remarkable agents and helpful for molecular and synthetic biology. In this review, we describe how geminivirus-based vectors can be utilized to overcome traditional transgenesis. We highlight the progress that has been made so far and also discuss the hurdles that hinder the employment of geminivirus-based vectors. Furthermore, we conclude with a comparison of geminivirus-based vectors with other plant-derived vectors. Geminivirus-based vectors stand poised to revolutionize plant genome editing by making nucleic acid manipulation cheaper and easier to deploy, thus lessening the major technical constraints, including homology-directed repair (HDR)-mediated genome editing and time-inefficient tissue culture procedures. The insights given in this review illustrate a broader picture of geminiviral vectors, with an emphasis on engineering plant viruses to ease genome editing practices for crop improvements as well as boost experimental timescales from years to months.

## 1. Introduction

Global agricultural productivity faces a great threat from plant viruses, which lead to significant economic losses and food insecurity. According to the Food and Agriculture Organization (FAO), nearly 40% of global agricultural production is lost due to diseases and plant pest attacks [1], with plant viruses contributing significantly. Among the plant viruses, geminiviruses (family: *Geminiviridae*) are infamous for their wide host range, efficient vector transmission, and ability to cause severe diseases in many economically important crops, including cotton, wheat, tomatoes, cucumbers, and papayas [2]. Geminiviruses get their name from their unique twin-icosahedral capsid structure. These viruses have a small, circular, single-stranded (ss) DNA genome, which is around 2.5 to 5.5 kb in length, encoding the proteins essential for multiplication and transmission in the host plants (reviewed in [3,4]). The destruction caused by geminiviruses is well documented. For example, the tomato yellow leaf curl virus (TYLCV), vectored by whiteflies (*Bemisia tabaci*), has severely impacted the production of tomatoes across 40 countries [5]. Cassava mosaic disease (CMD), caused by cassava mosaic geminiviruses (CMGs), jeopardized food security for millions of farmers as it led to 80% yield losses in Sub-Saharan Africa [6]. Similarly, the cotton industry of the Indian subcontinent has suffered losses amounting to billions of dollars due to cotton leaf curl disease (CLCuD) caused by the cotton leaf curl viruses (CLCuVs), which have been responsible for multiple epidemics in the region [7,8]. One reason for the appearance of epidemics in that region is the dispersal of a single species of whiteflies, *B. tabaci* (Asia II 1) [9]. The virus–host interaction is usually manifested by the development of symptoms, including vein thickening, plant stunting, leaf curling, and chlorosis. All these symptoms exhibit strong viral hijacking and a limited host armory. This phenomenon provokes the idea that geminiviruses have the capacity to deliver heterologous genes into plants and can provide help in altering plant genomes for trait improvement.

Plant virus-derived vectors have been widely utilized in plant genome engineering (reviewed in [10,11]) for efficient gene targeting (GT) and precise gene insertions through homologous recombination (HR) [12]. The main goal for efficient genome editing is to improve plant traits to cope with major issues arising due to climate change, increased pest pressures, and abiotic stresses. Among the genome editing technologies, the clustered regularly interspaced palindromic repeat (CRISPR)-/CRISPR-associated nuclease 9 (Cas9) system offers a precise and cost-effective method to target plant genes by inducing mutations. Recently, the potential application of geminivirus-derived vectors has gained much attention in targeting specific DNA as well as integrating donor DNA [13,14,15], thus opening a world of opportunities by expanding efficient genome editing possibilities.

The aims of this review are as follows: (1) to provide an understanding about the biological characteristics of geminiviruses, particularly those that can be utilized as vectors for genome editing (for instance, begomoviruses and mastreviruses); (2) to give an overview of engineering geminivirus-based vectors and their implications in plant trait improvement; (3) to provide instances of successful geminivirus-based vectors and their future prospects; (4) to compare DNA viral vectors with RNA viral vectors; and (5) to highlight the regulatory frameworks and consumer concerns regarding genome editing crops.

## 2. Harnessing Virus Engineering for Advanced Crop Breeding

Viral vectors are emerging tools in agriculture for delivering nucleic acids, proteins, and/or peptides to fine-tune crucial plant traits. Based upon successful evidence in human and veterinary medicine, viral vectors offer promising and flexible DNA-free approaches to crop improvement by utilizing both transient and heritable reprogramming in crop plants. Viral vectors offer a swift fine-tuning of crop traits with precision and can alter plant gene expression, metabolism, and epigenetic make-up in intact plants. For instance, using a viral vector delivery system to express flowering locus T (FT) can trigger on-demand flowering in crops like tomatoes, peppers, and cucumbers. This approach has also been employed to accelerate breeding in cotton, citrus, grapevine, and apple plants [16]. Plant architecture, another vital agronomic trait, can be modified by targeting hormones such as gibberellin (GA). Transient viral expression of GA-degrading enzyme GA2ox has led to more compact plant growth in tomato and pepper. Similarly, the viral delivery of transcription factors like DREB1A and ANT1 has enhanced drought tolerance and boosted health-related metabolites in tomatoes [17]. Apart from intermittent effects, viral vectors can mediate heritable characteristic change. In wheat, guide RNAs (gRNAs) delivered by viruses into Cas9-expressing lines offer specific changes to GASR7 genes, hence affecting grain size and weight [18]. When crossed with non-transgenic plants, they result in virus-free, genome-edited wheat lacking any foreign DNA. A similar approach in tomatoes targets the STAYGREEN1 (*SGR1*) gene, therefore enabling consistent changes that alter fruit color, i.e., a key quality for market appeal [19].

Viral vectors’ speed, accuracy, and adaptability make them a key tool for future agricultural developments. Targeting underutilized nutrient-rich crops with biocontainment indicators and blending them with local breeding programs could improve local food security and promote sustainable agriculture.

## 3. Geminiviruses and Their Unique Characteristics

Geminiviruses are the second-largest family of plant viruses. Several factors, including their genomic organization, wide host range, insect vector, and tissue tropism, make these viruses unique. Geminiviruses encompass fifteen genera: *Becurtovirus*, *Begomovirus*, *Capulavirus*, *Citlodavirus*, *Curtovirus*, *Eragrovirus*, *Grablovirus*, *Maldovirus*, *Mastrevirus*, *Mulcrilevirus*, *Opunvirus*, *Topilevirus*, *Topocuvirus,* and *Turncurtovirus* [20]. All these genera collectively contain 549 species, which cause important diseases (both in monocots and dicots) in most tropical and subtropical regions around the world. Among these genera, *Begomovirus* (464 species), *Curtovirus* (3 species), and *Mastrevirus* (50 species) comprise the majority of viral species. Mastreviruses and curtoviruses possess monopartite (single genome component) genomes, are vectored by leafhoppers, and cause diseases in vegetable and cereal crops [21]. In contrast, begomoviruses, which may have either one (monopartite/DNA-A) or two (bipartite/DNA-A and DNA-B) genome components, are primarily transmitted by whiteflies and infect vegetable, root, and fiber crops [22] (Figure 1). Furthermore, begomoviruses are also reported to be associated with circular, single-stranded DNA satellite molecules, known as alphasatellites (previously known as DNA-1; ~1.3 kb), betasatellites (formerly known as DNA-β; ~1.3 kb), and deltasatellites (~0.7 kb) [23]. Curtoviruses and monopartite begomoviruses share a similar genomic organization, consisting of a single intergenic region (IR) that contains bidirectional promoters for viral gene expression and the origin of replication. They encode two or three proteins (V1, V2, and V3) on the virion (V) strand and four or five proteins (C1, C2, C3, C4, and C5) on the complementary (C) strand. Furthermore, bipartite begomoviruses contain a single IR and encode two proteins, one on the V-strand, which encodes the nuclear shuttle protein (NSP; BV1), and another on the C-strand, which encodes the movement protein (MP; BC1) [24]. Broadly, begomoviruses can be divided into four phylogenetic groups: (1) Old World; (2) New World; (3) sweepoviruses, and (4) legumoviruses. The Old World begomoviruses include both monopartite and bipartite viruses and typically possess the V2 protein, whereas New World begomoviruses are mostly bipartite and lack the V2 protein [25,26]. They share a common region (CR) within the IR (also referred to as the long IR; LIR). On the other hand, mastreviruses possess two IRs: a long IR and a short IR (SIR). These viruses utilize transcript splicing, and thus their genomes contain introns. The C1 and C2 open-reading frames (ORFs), involved in replication, encode the replication initiator protein. Additionally, the V-strand encodes the V2 and coat protein (CP).

## 4. Geminiviruses: Master of the Host Genome’s Manipulator

Geminiviruses, with their small genome size, chiefly rely on host machinery for replication and symptom progression. Due to their limited coding capacity, these viruses encode four to eight multifunctional proteins, including the newly identified small additional proteins necessary for infection and replication [24,27]. Geminivirus proteins target a plethora of highly connected proteins (hubs) of the plant cell to redirect the cellular functions for viral genome replication and spread [28]. This virus–host interaction leads to altered gene expression, suppression of cell death mechanisms, interference with plant signaling pathways, and suppression of plant defenses, including the inhibition of antiviral silencing pathways [29,30].

### Geminiviruses Lifecycle: Hijacking Host Nuclear Machinery

Geminiviruses begin their journey by entering and hijacking plant cell nuclei. Their ssDNA genome replicates in the nucleus through rolling circle replication (RCR), which first involves the conversion of the ssDNA genome of the virus into double-stranded (ds) DNA with the help of plant DNA polymerase. This dsDNA molecule acts as a template for the replication and transcription of viral genes. The geminivirus-encoded (Rep/RepA/C1) protein plays a pivotal role in initiating host cell reprogramming by interacting with several host cellular proteins. Then, the Rep/RepA identifies a nonanucleotide sequence within the conserved IR of the viral DNA and produces a site-specific nick on one of the two strands of the dsDNA molecule. Consequently, by utilizing host DNA polymerase, Rep initiates replication and mediates circularization, followed by the release of a nascent virion (ssDNA) molecule. Additionally, replication enhancer protein (Ren/C3) collaborates with Rep/RepA along with other host replication-associated proteins to boost the accumulation of viral genomes [31]. Eventually, newly formed viral molecules move inter-/intracellularly and invade new host cells to generate new ssDNA molecules available for acquisition by the insect vector. Viral DNA replication has also been modulated by a recombination-dependent replication process, which is based on homologous recombination [4]. However, Rep/RepA is sufficient for the effective replication of the viral genome.

After winning the battle in the nucleus and conquering the plant antiviral defenses, geminiviruses cross cellular boundaries, move from cell to cell, and establish systemic infections. Within the cell, CP transfers viral DNA into the nucleus, followed by the uncoating and generation of new virions. The newly synthesized ssDNA or dsDNA molecules are transferred from the nucleus to the cytoplasm by NSP in bipartite geminiviruses and CP in monopartite geminiviruses. Reaching the cellular periphery, plasmodesmata (PD) are reportedly involved in facilitating the cell-to-cell movement of the viral genomes. Bipartite viruses encoded MP found at PD openings, facilitating this passage by pushing the exclusion limits of PD. In monopartite geminiviruses, similar roles have been speculated for the C4 protein [32]. Understanding the roles of viral components is essential for the effective utilization of geminiviruses in plant biotechnology. The Rep and IR regions are the most vital elements required for geminivirus replication. Efficient replication of a vector system is the most desired feature in molecular biology; however, cell-to-cell movement and encapsidation—typical features of the viral life cycle—are the least desirable in viral vectors [33]. Thus, this presents an opportunity to customize viral genomes for the development of geminivirus-derived vectors.

## 5. Geminiviruses: How They Can Help in Plant Genome Editing

### 5.1. Overview of CRISPR-Cas System

In the field of plant genetic engineering, manipulating the plant genome to enhance production or improve resistance to abiotic and biotic stresses is fundamental. DNA manipulation is achieved through various genome-editing technologies, which can be broadly categorized into two groups: (1) conventional genome editing technologies, including meganucleases, zinc-finger nucleases (ZFNs), and transcription activator-like effector nucleases (TALENs); and (2) clustered, regularly interspaced short palindromic repeats-CRISPR/CRISPR-associated proteins (CRISPR-Cas), which offer greater efficiency and precision compared to conventional approaches. There are three key qualities that make CRISPR-Cas system the most efficient tool for genome editing: (A) high sequence specificity; (B) ease of programmability; and (C) capability of adding custom functionalities including reverse transcriptase (prime editors), deaminases (base editors), and transcriptional repressor and enhancer (for gene regulation). The mechanism of the CRISPR-Cas system involves nucleases, such as Cas9 and a guide RNA (gRNA); together, they induce double-stranded breaks (DSBs) in the DNA, which are typically repaired through the non-homologous end joining (NHEJ) pathway, resulting in small insertions or deletions indels (Figure 2). In plants, the gRNA-Cas9 complex can induce various genome structural variants, including deletions, insertions, inversions, and even inter-chromosomal translocations. However, this approach is limited in its ability to edit large DNA fragments. Alternatively, the homology-directed repair (HDR) pathway can be utilized, wherein a donor DNA template is introduced to enable the editing of large DNA fragments. Nonetheless, the HDR-based genome editing remains highly challenging and inefficient in plants, primarily due to the low incorporation efficiency of the donor DNA template into the plant cells [34]. Based on the editing outcomes, site-directed nucleases (SDNs) are divided into three types: (1) SDN-1 involves genome editing without the use of a DNA template and typically results in small indels. The indels are generated through the NHEJ repair pathway and can potentially lead to gene knockout by disrupting the gene sequence [35]; (2) SDN-2 utilizes a short donor DNA template that is homologous to the target region and enables the introduction of small insertions, deletions, or single-nucleotide variant (SNV) through the homology-directed repair (HDR) pathway; (3) SDN-3 involves the insertion of a large DNA template (kilobase-scale) into a specific genomic locus, also via HDR, to the cleavage site. SDN-2 and SDN-3 rely on the HDR-repair mechanism, thereby allowing precise allele replacement, gene modification, or the insertion of new coding sequences (Figure 2). In general, CRISPR-based genome editing encompasses several approaches: nuclease editing, which generates small indels, base and prime editing, which enable precise base substitutions and small indels; and large DNA fragment editing, which employs advanced strategies such as prime editing, CRISPR-associated recombinases, or transposases to insert or modify kilobase-scale sequences [36,37,38]. HDR-driven knock-in (KI) remains very challenging due to the inefficient delivery of the repair template (i.e., donor DNA). However, several strategies have been employed to enhance HDR frequency, including the overexpression of genes involved in homologous recombination (HR), the use of single-stranded donor DNA, and the delivery of donor template via viral vectors such as DNA or RNA viruses [39]. To date, *Agrobacterium*-, biolistic-, floral dip-, and polyethylene glycol (PEG)-mediated transformation methods are extensively used for DNA delivery in plants. However, these methods have several limitations, including species dependency, embryonic mortality, high cost, time inefficiency, and low transformation efficiency. Alternatively, plant viruses, including RNA and DNA viruses, are being explored as vectors for delivering CRISPR reagents, offering a promising solution (reviewed in [10,11,40]. The ability of viruses to autonomously replicate within plant cells and produce a high copy number of donor DNA leads to enhanced protein expression, making them well-suited for efficient cargo delivery.

### 5.2. Geminivirus-Based Vectors: A Stalwart Approach for Genome Editing

Over the last 40 years, geminiviruses have been exploited to deliver CRISPR reagents and repair templates of varying length into plant cells. Geminiviral DNA replication primarily depends on two viral elements: Rep/RepA and the IR. These two elements serve as the building blocks for replicon replication in plant cells. The sequence of interest is typically flanked by two IRs (or LIRs in case of mastreviruses) (Figure 3a). Upon expression of the Rep/RepA protein, DNA molecules comprising the gene of interest are produced. Notably, the production of replicons is independent of whether Rep/RepA is expressed from the viral genome or introduced transgenically. These molecules replicate due to the activity of Rep/RepA and accumulate abundantly within the cell. This high replication efficiency makes geminiviruses particularly well suited for HDR-based genome editing, which relies on the presence of donor DNA with a high copy number. By following this ability of geminivirus-derived replicons, Gil-Humanes et. al. and Wang et al. completed successful insertions in wheat and rice genomes [13,41], which is discussed later in this review.

The short and long intergenic regions (SIR and LIR, respectively) in the geminiviral genome enable a linear construct to circularize through the joining of two LIR sequences, facilitated by the viral Rep/RepA proteins (Figure 3b). This allows employing ‘LSL’ (LIR-SIR-LIR) vectors, in which a linear construct—such as a deconstructed BeYDV vector (discussed in detail in the following section) (Figure 3c)—containing an SIR flanked by two LIR can be introduced into an *Agrobacterium* strain, followed by *Agrobacterium*-mediated transformation of the plant (Figure 4) [42]. Broadly, geminiviruses can be modified and utilized for nucleic acid delivery through two main strategies: (1) the full virus vector strategy and (2) the deconstructed virus vector strategy [43].

### 5.3. Full and Deconstructed Virus Vector Strategy

In the full-virus vector strategy, geminiviruses keep most of the genes necessary for replication and host infection. The gene of interest (heterologous sequence) can be inserted into the virus genome by replacing the *CP* gene. In some bipartite geminiviruses, the CP is dispensable for cell-to-cell movement (Figure 3c), making it suitable for such modifications [44]. In a study by Hyes et al., a bipartite geminivirus (tomato golden mosaic virus; TGMV) was used to express the neomycin phosphotransferase (*neo*) gene in *N. tabacum* cv. Samsun [45]. The authors developed a chimeric DNA by replacing the viral *CP* gene with *neo* and demonstrated that the modified virus was able to replicate and spread systemically throughout the plant. Notably, the study confirmed that CP is not required for successful infection or systemic spread in this case [46]. Another example of a DNA virus deployed as a gene replacement vector is cassava latent virus (CLV). The CLV *CP* gene was replaced with the bacterial chloramphenicol acetyl transferase (*CAT*) gene. The *CAT* gene was expressed under the control of the CP promoter, and the study exhibited successful systemic expression in *N. benthamiana* leaves, with CAT activity reaching 80 U/mg [47]. Furthermore, Baltes et al. utilized a recombinant cabbage leaf curl virus (CaLCuV) to deliver a repair template. They targeted the *Arabidopsis thaliana* alcohol dehydrogenase (*ADH1*) gene. The authors modified the DNA-A component of the CaLCuV genome by incorporating an 18-bp sequence into the repair template (the total size of the donor DNA was 600 bp). After infecting the *Arabidopsis* plants with CaLCuV, the authors identified a single plant with the desired insertion, suggesting that the full virus vector can be effectively deployed for precise gene insertion [15].

A disadvantage of this strategy is that integrating large heterologous sequences, up to 800 bp, may not be feasible [48]. The limitation in genome size is due to constraints in cell-to-cell movement via plasmodesmata, rather than replication capacity [49]. However, this capacity is insufficient for expressing TALENs and Cas9; it is suitable for expressing gRNAs. As an alternative approach, researchers have generated stable Cas9 transgenic plants and used viral replicons to express gRNAs [50], thus eliminating size limitations and enabling targeted genome engineering.

In the deconstructed virus vector strategy, undesired components (e.g., MP and CP) are removed, and only useful components [e.g., *cis-*(IR) and *trans-*(Rep/RepA)], which are crucial for replication, are retained (Figure 3c). The movement requirement is fulfilled by using *Agrobacterium*-mediated transformation to deliver the viral vector into plant cells.

### 5.4. BeYDV: A Successful and Pioneer Geminivirus-Based Vector

Several successful applications of utilizing deconstructed viral replicons have been reported. For instance, one of the pioneering geminivirus-based vectors, bean yellow dwarf virus (BeYDV), was deployed to express vaccine proteins, reporter proteins, and monoclonal antibodies [42]. The deconstructed virus vector strategy offers several advantages over the use of the full virus vector strategy. For example, no cargo size limit is known with this strategy. However, replication efficiency may be affected as the replicon size increases, and the potential lethal effects associated with other viral proteins are typically overlooked. Moreover, this strategy imposes fewer restrictions related to host range, though this aspect still requires careful consideration. A comprehensive list of deconstructed geminiviruses-derived replicons is provided in Appendix A.

Geminivirus-derived deconstructed vectors, including BeYDV, were used to efficiently deliver large sequences, including repair templates, ZFNs, and TALENs, to induce mutations in potato [51], tomato [14], and tobacco [15]. These replicons exhibit 10- to 100-fold higher efficiency compared to traditional *Agrobacterium* T-DNA delivery. The BeYDV replicon system was also utilized to express *Streptococcus pyogenes* Cas9 (SpCas9). Furthermore, BeYDV replicons were also utilized for targeted mutations in potato and cassava to enhance herbicide tolerance [51,52]. Beyond gene targeting, geminiviral vectors can replicate inside the plant cell nucleus with high copy numbers, thus unveiling the utilization of geminiviral replicons as a repair template for HDR. By merging this replicon strategy with *Agrobacterium*-mediated T-DNA, it would be possible to obtain high editing outcomes in a more efficient manner [53] (Figure 4). One way to utilize this strategy is by loading the repair template onto a geminiviral replicon, while nucleases can be delivered either via the replicon or through traditional T-DNA delivery methods. A notable example of employing BeYDV is an HDR-based genome editing in tobacco and tomato, where the replicons successfully restored the precise wild-type sequence using a donor template and incorporated a strong 35S promoter upstream of the tomato endogenous *ANT1* gene. The method yielded an efficiency that was ~12 times higher than standard *Agrobacterium* T-DNA delivery [14]. BeYDV-derived vectors have been effectively utilized to investigate the effects of phytohormones on plant biomass using a transient expression system, and efficiency was recorded at 19–25% higher than controls for different hormones [54]. The aforementioned example highlights the adaptability of geminiviral replicons, indicating that these replicons can be modified and merged with other delivery methods, such as T-DNA, nanoparticles, etc. We provide a curated list of geminiviruses that have been deployed as vectors for genome editing, along with detailed descriptions, in Table 1.

### 5.5. Geminivirus-Based Homology-Directed Repair: Case Studies

Two geminiviruses, wheat dwarf virus (WDV) and tomato leaf curl virus (ToLCV), were used for gene targeting in wheat, corn, and rice [13,41]. In a study by Gil-Humanes et. al., a deconstructed version of wheat dwarf virus (WDV) expressed the reporter gene in the wheat cells at a 110-fold increase over the control. Gene targeting efficiency when delivering Cas9 and repair template in a ubiquitin locus resulted in 12-fold higher efficiency than non-viral methods. The study also highlighted that the nuclease (Cas9) promoter played a significant role in efficient GT by comparing the maize ubiquitin promoter (ZmUbi) with the viral LIR promoter and suggested that ZmUbi significantly enhances the GT. The findings of targeted integration by HDR in all three wheat homoeoalleles (A, b, and D) [13] provide evidence that geminiviral-based replicons have the potential for precise and efficient HR-based genome editing in cereals.

In another study by Wang et al., geminiviral-based knock-in (KI) in rice was executed by using WDV. They selected two loci (*ACT1* and *GST*) for targeted KI and incorporated GFP-fused cassettes (ACT1-GFP and GST-GFP). The cassettes were inserted into WDV replicon to make WDV expression cassettes (WDV1-ACT1 and WDV2-GST, respectively), followed by the incorporation of U6-driven gRNA. One barrier in the study was the larger Cas9 size, which prevented them from being able to incorporate into the replicon. To overcome this, they generated transgenic rice calli expressing Cas9, which were used to inoculate WDV vectors. They found that the targeted KI frequency for *ACT1* and *GST* was 8.5% and 4.7%, respectively, which was lower than using Cas9-expressing rice calli [41].

Another geminivirus-based vector, Beet curly top virus (BCTV), was deployed to deliver Cas12a in GFP-transgenic *N. benthamiana*. The virion-sense genes (V1, V2, and V3) were removed, and a constitutive cauliflower mosaic virus (CaMV) 35S promoter was inserted upstream of the Cas12a. They found a higher indel rate of approximately 40% in the GFP target region compared to *Agrobacterium*-mediated T-DNA delivery. Furthermore, they compared three different Cas nucleases (LbCas12a, AsCas9, and SpCas9) and found that the LbCas12a nuclease showed higher efficiency with no evidence of off-target mutations [55]. Moreover, BCTV also represented a higher rate of mutagenesis than previously deployed geminiviruses, including BeYDV and sweet potato leaf curl virus (SPLCV) [15,56].

Sarrion-Perdigones and team optimized the BCTV-based viral vector system using the GoldenBraid (GB) modular cloning platform, which employs standardized DN parts with Type IIS restriction enzymes and 4-nt overhangs [57]. Furthermore, it was also found that a native BCTV promoter inclusion upstream of the gene of interest (GOI) performed better than the previously used constitutive CaMV 35S promoter. Moreover, BCTV-replicons also accommodated a cargo of up to 4 kb long without impacting replication. Furthermore, removal of virion-sense genes further enhanced gene expression.

Over the past decade, several advancements in the field of genome editing and omics have provided much evidence for utilizing geminiviral vector technology [37,58,59]. Furthermore, HDR-based genome editing, once a major challenge, is now achievable through the power of geminivirus vectors. Nevertheless, just like every past technology, virus-induced genome editing (VIGE) has limitations, including off-targets, limited cargo capacity, elimination of viruses in systemically infected plants, heritable editing, the fact that germline cells must be infected, etc.

Recently, to reduce CRISPR-mediated off-targets, Vu et al. utilized a novel combination of prime editing (PE) components, which consists of RNA chaperone and modified prime editing guide RNAs along with a geminiviral (BeYDV)-based replicon system, and obtained 9.7% desired PE efficiency at the callus stage. They carried out this experiment in tomatoes and *Arabidopsis* and obtained 38.2% positive transformants that contained desired PE alleles [60], proving potentially heritable PE transmission and unlocking the practical applications of geminiviral replicons in dicots. Another efficient VIGE strategy to obtain genome-edited progeny involves delivering only gRNA via geminiviral replicon into a stable transgenic plant expressing Cas9. This method has emerged as a powerful tool in plant genome engineering. In a study by Yin et al., the authors engineered a cabbage leaf curl virus (CaLCuV) to deliver gRNA by replacing its CP with gRNA [50]. Unlike conventional virus-induced gene silencing (VIGS), geminiviral replicon-based genome editing (VIGE) has two advantages. First, VIGS can cause additional non-specific silencing, especially for homologous genes, while CRISPR-based VIGE can only target specific genes and cause knock-out after repairing DSB by NHEJ, providing an optimal tool to study specific gene function. Second, cloning is required to synthesize a fragment of a targeted gene by PCR in VIGS. Only a small fragment of about 20 bp in length of the target sequence is required to be put into gRNA, providing a simple and ideal way to establish a high-throughput genome-wide functional analysis.

**Table 1 viruses-17-00631-t001:** List of geminivirus-based vectors used in genome editing.

Viruses	Genus	GE Platform	gRNA Type	Plant Species	Targeted Gene *	Heritability	Reference
BeYDV	Mastrevirus	CRISPR-Cas9	AtU6-gRNA	*Solanum tuberosum*	*StALS1*	Yes	[61]
		TALEN and CRISPR-Cas9	AtU6-gRNA	*S. lycopersicum*	*ALS1*	No	[51]
		ZFN, TALEN and CRISPR-Cas9	AtU6-gRNA	*N. tabacum*	*ALS* and *P-GUS: NPTII*	No	[15]
		TALEN and CRISPR-Cas9	AtU6-gRNAs	Tomato cv. MicroTom	*ANT1*	Yes	[14]
		CRISPR-Cas9	AtU6-gRNAs	*Lycopersicon esculentum*	*CRTISO* and *PSY1*	Yes	[62]
WDV		CRISPR-Cas9	TaU6-gRNA	*Triticum aestivum*	*Ubi*, *MLO*, and *GFP*	No	[13]
		Cas9 expressing rice	OsU6-gRNAs	*Oryza sativa*	*ACT1* and *GST*	No	[41]
CaLCuV	Begomovirus	Cas9 expressing tobacco	AtU6-gRNA	*N. benthamiana*	*NbPDS3* and *NbIspH*	No	[50]
SPLCV		LwaCas13, LbCas12a and Cas9	AtU6-gRNA	*N. benthamiana*	*NbPDS1* and *mGFP5*	No	[56]
BCTV	Curtovirus	CRISPR/Cas12a	-	*N. benthamiana*	*NbGFP*	Yes	[55]

* *Solanum tuberosum* acetolactate synthase 1 (*StALS1*), acetolactate synthase 1 (ALS1), promoter of GUS and neomycin phosphotransferase II (P-GUS: NPTII), anthocyanin mutant 1 (ANT1), carotenoid isomerase (CRITSO), phytoene synthase 1 (PSY1) ubiquitin (Ubi), Mildew Locus O (MLO), green fluorescent protein (GFP), Actin-1 (ACT1), glutathione S-transferase (GST), *Nicotiana benthamiana* phytoene desaturase 3 (NbPDS3), *Nicotiana benthamiana* isopentenyl/dimethylallyl diphosphate synthase (IDDS).

## 6. Pros and Cons of Geminivirus-Based Vectors

The circular ssDNA genome of geminiviruses has certain advantages and disadvantages over linear RNA viruses—for instance, the cargo capacity, genomic size, organization, and ability to manipulate host genetic material. We list a few points based on the research that has been made so far and provide a few potential strategies that could help in the utilization of plant viral vectors. This includes the following factors: (i) The ability of geminiviruses to autonomously replicate within the host nucleus at high levels, which has been the key to their success as vectors. To date, HDR-based genome editing remains highly challenging. However, recent studies have demonstrated the feasibility of using geminivirus-based vectors to deliver donor DNA in plants [15,41,55]. Thus, geminiviruses can easily integrate heterologous proteins in their circular genome and can offer potential to stably deliver nucleases, gRNAs, and donor DNA to the linear genome of RNA viruses. (ii) The size constraints of geminiviruses can be negotiated by removing unnecessary viral genes, including coat and movement proteins, a strategy known as the deconstructed viral vector (Figure 3c) [13,14], thus providing huge potential to facilitate larger proteins in geminivirus-based vectors. In addition to removing virion-sense genes, various heterologous promoters have also been utilized. Eini et al. successfully delivered Cas12a (Cpf1; ~1300 amino acids) into plants by removing the virion-sense genes of the Beet curly top virus (BCTV). To express Cas12a and other cargoes, they employed the CaMV 35S promoter [55]. A BCTV-based vector can accommodate cargo up to 4 kb in size without affecting viral replication [33]. (iii) Newly discovered smaller-sized Cas nucleases can provide a way forward to precisely and efficiently screening the gene function of plants. The smaller Cas nucleases include *Campylobacter jejuni* Cas9 (CjCas9; 984 amino acids), CasΦ (Cas12j; 800 amino acids), and more recently discovered NanoCas (425 amino acids) [63,64,65]. It is important to note that NanoCas has not yet been tested in plants. However, if it demonstrates equal or higher efficiency than traditional Cas9, it could be a game-changer for plant biotechnology. This would allow the use of a single viral vector instead of generating two viral cassettes to express Cas and gRNAs simultaneously. (iv) RNA-viruses like Sonchus yellow net rhabdovirus (SYNV) lack a DNA phase during replication, and due to this ability, RNA-virus-based vectors have the advantage of not integrating into the host genome. Thus, mutant plants regenerated from infected tissues are non-transgenic. On the contrary, it is extremely difficult to eliminate geminivirus-based vectors via regeneration, as geminiviral proteins interact with various host cellular proteins for viral replication. A plausible solution to this would be the modification of strictly inducible geminivirus vectors that can only express when the plant reaches a specific biomass. (v) One of the major disadvantages of using geminiviral vectors is their perturbation of the plant cell cycle during the viral replication process [66]. Meanwhile, RNA viruses do not have this drawback, as the genomes of these viruses are packaged within their capsids.

## 7. Outlook of Viral Vector-Based Crop Design

Deploying plant virus-based vectors to provide heterologous proteins in plants offers an effective and time-efficient procedure. To date, a number of RNA viruses, including TRV, bamboo mosaic virus (BMV), BYSMV, BSMV, beet necrotic yellow vein virus (BNYVV), pea early browning virus (PEBV), PVX, and SYNV, have successfully demonstrated targeted and heritable mutations [67,68,69,70,71] in dicot plants, whereas a number of viral vectors, including WDV, BSMV, and foxtail mosaic virus (FoMV), have been reported in monocot plants [13,72,73,74]. Apart from this, a number of RNA viruses, including BaMV, BSMV, BYSMV, brome mosaic virus (BMV), cymbidium mosaic virus (CymMV), cucumber mosaic virus (CMV), FoMV, rice tungro bacilliform virus (RTBV), and wheat streak mosaic virus (WSMV), are deployed for gene suppression and systemic protein expression analysis [73,75,76,77,78,79,80,81]. Once the editing efficiency and heritable mutations of modified alleles are optimized, VIGE would bypass the need for plant regeneration and transformation, thus boosting the breeding of new climate-friendly crop varieties. Moreover, the absence of exogenous sequences (in the case of RNA virus-derived vectors) would make genome editing indistinguishable from the traditional mutagenesis approaches and make regulatory approval easy [82].

Plant viral vectors can be utilized to rewire different biosynthetic pathways by transferring transcription factors, precise and targeted knock-down or overexpression of metabolic genes, or heterologous enzymes. Similar methods might be applied for field-grown crops biofortification as an alternative or in conjunction with traditional breeding and transgenic approaches. Furthermore, viral vectors can be used to express positive regulators ectopically, as discussed before with an example of the *FT* gene [68]. Virus-induced flowering enhances *FT* accumulation; early flowering; and has further been applied to promote breeding programmes and genetic studies of cotton, citrus, and apple [83]. In addition, virus-induced flowering is also helpful for tree species with a long juvenile phase that can take years or even decades.

Geminiviruses have a wide host range that makes them unprecedented in the development of viral vectors for delivering genome-editing reagents. With the discovery of miniature Cas (NanoCas), it would be much easier to manage multiple gRNAs along with a nuclease for precise and efficient gene targeting. Multipartite viruses, including geminiviruses (helper viruses/satellites), can also be engineered for multiple gene silencing and protein co-expression assays, as suggested by Liou et al. [84]. Moreover, geminiviral vectors are potentially useful for efficient genome editing, especially in transient assays in model plants such as tobacco, as well as in the regeneration of transgenic plants harbouring the desired mutations. As geminiviruses are not seed-transmissible, deconstructed geminiviral strategies could potentially be utilized for genome editing in plants, offering a novel system to obtain stable mutant progeny in a single generation, for example, by modifying reproductive tissues.

Lastly, regulatory frameworks and consumer concerns regarding genome-edited crops and the use of viral vectors cannot be overlooked. Globally, consumers are showing misconceptions, inadequate understanding, and a general unfamiliarity with genome editing technologies in plants [85]. Such consumer behavior is influenced by cultural, social, personal, psychological, and economic factors [86]. Nonetheless, without embracing these ground-breaking technologies, the issues of genome editing and, hence, global food security, will be a monumental challenge.

## Figures and Tables

**Figure 1 viruses-17-00631-f001:**
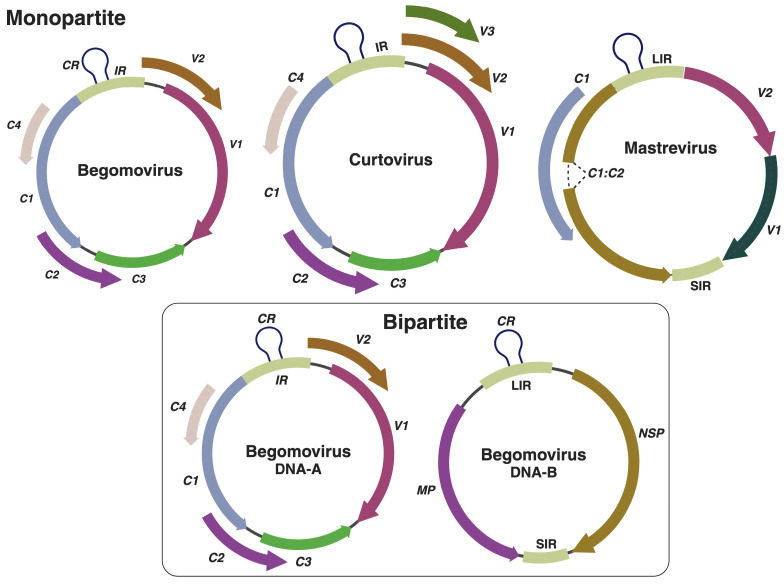
Genomic organization of begomoviruses, curtoviruses, and mastreviruses. CR: common region; IR: intergenic region; LIR: long intergenic region; SIR: short intergenic region; NSP: nuclear shuttle protein; MP: movement protein. Created in https://BioRender.com.

**Figure 2 viruses-17-00631-f002:**
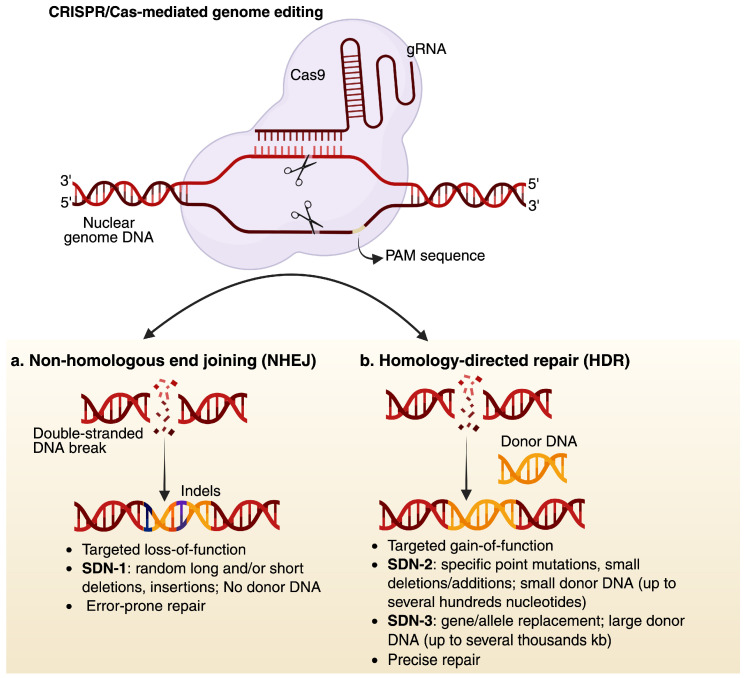
CRISPR-Cas9 system and its types. The basic working principle of CRISPR-Cas9 system. This system works on a simple DNA-RNA base pairing, the PAM sequence, and target-specific gene precisely. Upon targeting, it results in DNA DSBs, which are repaired either with NHEJ, which is an error-prone process and results in random indels and gene disruption at the cleavage site (termed as SDN-1 process), or with HDR, which precisely repairs the DSBs. HDR process can be harnessed to integrate specific DNA template at the cleavage site (termed SDN-2 and 3). The donor template could be several hundreds to thousands of base pairs long. PAM: protospacer adjacent motif. Created in https://BioRender.com.

**Figure 3 viruses-17-00631-f003:**
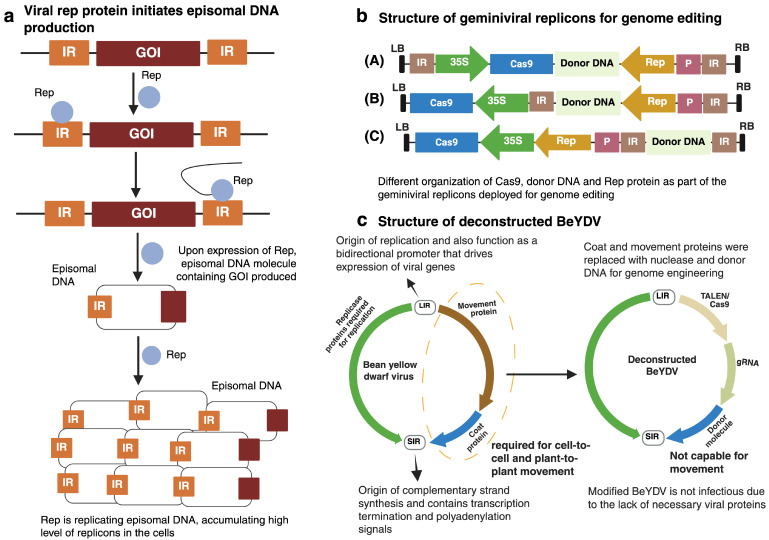
Geminivirus-based replicon structure. (**a**) Basic geminiviral-based replicon mechanism. The gene of interest is flanked by two direct IRs. Upon expression of Rep, episomal replicons are generated, thus accumulating at high levels in the cell. (**b**) Geminiviral replicon structures for genome editing. Three versions of modified cassettes are shown here. In (A), the Cas9, donor DNA and Rep protein are included as part of the replicon; in (B), the donor DNA and Rep protein are part of the replicon, while Cas9 is outside the replicon and is only expressed from T-DNA; in (C), only the donor DNA is part of the replicon, while the Cas9 and Rep protein are outside the replicon. Note that the Rep protein does not need to be part of same T-DNA but can also be provided independently. Replicons such as (A) and (B) have been successfully utilized in genome editing [15]. (**c**) The genomic structure of bean yellow dwarf virus (BeYDV) and modified BeYDV. LB: left border, RB: right border, 35S: cauliflower mosaic virus (CaMV) 35S promoter, and P: promoter. Created with https://BioRender.com.

**Figure 4 viruses-17-00631-f004:**
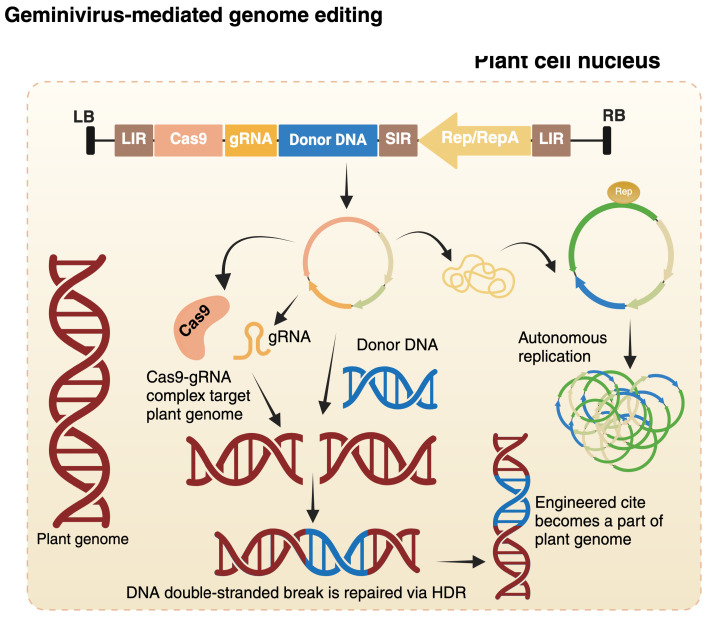
Illustration of geminivirus-mediated homology-directed repair (HDR) genome editing. The modified geminivirus genome containing Cas9, gRNA, and donor DNA is cloned into the transfer DNA (T-DNA) of *Agrobacterium*. During *Agrobacterium* infection, the T-DNA molecules are transferred to the plant cell nucleus, where the viral genome (circular) is amplified into thousands of copies via rolling circle replication driven by replicase proteins. On the other side, the complex of gRNA and Cas9 induces double-stranded breaks (DSBs) at the target locus, and the donor DNA is integrated into the target site via HDR pathway and, ultimately, the integrated donor DNA becomes a permanent part of the plant genome. Created with https://BioRender.com.

## Data Availability

No new data were generated.

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
