# Peer review of "Attaining the Promise of Geminivirus-Based Vectors in Plant Genome Editing"

_viruses, 2025, doi:10.3390/v17050631_

Round 1
Reviewer 1 Report
Comments and Suggestions for Authors
In their manuscript, Mahmood et al. provide a concise review of the application of geminiviruses in biotechnology, with a particular emphasis on the potential of geminivirus-based vectors for plant genome editing. Geminiviruses, constituting one of the largest families of plant viruses, are distinguished by their small, circular, single-stranded DNA genomes. While a similar review was published in 2016 (Lozano-Durán et al, 2016), this work offers some updated insights into the use of these viruses in biotechnological applications. The manuscript is generally well-organized and informative; however, it is marred by numerous writing errors, including grammatical inaccuracies, that detract from its overall quality. I recommend acceptance pending revisions to address the concerns outlined below, which should enhance its readability and scientific rigor.
Specific Concerns:
1.Page 1, Lines 19-20: The sentence "...this nature of viruses can be exploited for its best utilization in molecular biology" contains a subject-verb agreement error. The singular pronoun "its" is inconsistent with the plural noun "viruses." Additionally, "optimal" is a more precise and academic alternative to "best."
2.Page 1, Line 38: The family name "Geminiviridae" should be italicized.
3.Page 3, Lines 97-98: The statement "The V2 protein is not present in the DNA-B genome of the bipartite begomoviruses..." is inaccurate.
4.Page 10, Line 355: The sentence "Another strategy to obtained genome edited progeny is to deliver only gRNA with ......" is incomplete and contains a grammatical error ("obtained" should be "obtain").
5. The manuscript repeatedly discusses applications of the bean yellow dwarf virus (BeYDV) across multiple sections, leading to redundancy. To improve conciseness, I suggest consolidating all BeYDV-related content into a single, cohesive section and summarizing specific case studies in a table or supplementary material.
6. No supplementary file 1 can be found.
Comments on the Quality of English Language The authors should conduct a thorough proofreading to address pervasive grammatical issuesAuthor Response
We are thankful to the reviewer for their insightful comments and suggestions.
Comment 1: Page 1, Lines 19-20: The sentence "...this nature of viruses can be exploited for its best utilization in molecular biology" contains a subject-verb agreement error. The singular pronoun "its" is inconsistent with the plural noun "viruses." Additionally, "optimal" is a more precise and academic alternative to "best."
Response: The word ‘best’ is replaced by optimal and ‘its’ with ‘their’.
Comment 2: Page 1, Line 38: The family name "Geminiviridae" should be italicized.
Response: The suggested change has been implemented.
Comment 3: Page 3, Lines 97-98: The statement "The V2 protein is not present in the DNA-B genome of the bipartite begomoviruses..." is inaccurate.
Response: The statement has been modified and new referenced have also added.
Comment 4: Page 10, Line 355: The sentence "Another strategy to obtained genome edited progeny is to deliver only gRNA with ......" is incomplete and contains a grammatical error ("obtained" should be "obtain").
Response: The grammatical error has been corrected, and the sentence is now complete.
Comment 5: The manuscript repeatedly discusses applications of the bean yellow dwarf virus (BeYDV) across multiple sections, leading to redundancy. To improve conciseness, I suggest consolidating all BeYDV-related content into a single, cohesive section and summarizing specific case studies in a table or supplementary material.
Response: The section 4.4 has been added and all BeYDV studies have been gathered under one heading and rest of the summary is already available in table 1.
Comment 6: No supplementary file 1 can be found.
Response: Supplementary file information can be found in the last paragraph of 4.3.
Reviewer 2 Report
Comments and Suggestions for Authors
The review by Mahmood et al. is of interest to virologists and plant biologists. The MS is well structured, rather well-written. It contains the description of the main features and characteristics of geminiviruses and vectors for plant genome editing based on the genomes of these viruses. Also authors overview the examples of successful usage of geminivirus-based vectors for plant genes editing and, finally, discuss regulatory issues of this approach. Thus, the idea and the concept of the manuscript is interesting and could attract attention of the wide scientific audience.
However, there are several major concerns to be addressed:
Abstract should be re-written as now it doesn’t reflect the main idea and the content of the manuscript but contains some general words. It should be more precise and succinct. For example, the fragment about viruses: “Viruses being invisible infectious agents can severely impact on organism’s life cycle. They are obligate intracellular pathogens that have evolved efficient strategies to hijack and manipulate the host’s immune system, and this nature of viruses can be exploited for its best utilization in molecular biology.” is not necessary because the MS is submitted to the journal Viruses, thus the readers definitely acquainted with viruses in general.
The Introduction section should also be improved. It starts with the paragraph (L 34-37) telling that viruses are harmful for the agriculture – this statement is of course correct but doesn’t fit the main idea and the scope of the paper. In my opinion, Intro should contain information about viruses as a tool for biotechnology, viral vectors, geminiviruses’ general characteristics, very brief overview of methods used for editing and the significance of virus genome-based instruments for editing. The description of symptoms etc (L 44-56) is unnecessary and out of place here.
In general, I see lack of novelty and originality of the manuscript. The content partly repeats (not literally, of course) the review “Plant Virus-Derived Vectors for Plant Genome Engineering” by Mahmood et al., 2023 (looks like the same group) https://doi.org/10.3390/v15020531.
The section about RNA viruses does not fit the scope of the review, in my opinion, RNA viruses (as well as Gemini) have been already overviewed in the previous paper. Why did the authors include this large section in the review about geminiviruses? I think, that analysis of the advantages and limitation of the vectors based on the genomes of DNA viruses (Gemini) is enough and it could of course include the comparison with RNA viruses-based vectors, but the present description of these vectors looks superficial and unnecessary here in this review.
Minor:
- The acronym “HDR” should be introduced at the first use (in abstract and in the main text)
- L 68 – the sentence is incorrect
- L 114-115 – the sentence is not correct: Geminiviruses, with their small genome size, chiefly rely on host machinery for replication and symptom progression – while viral replication really depends on the cellular components, it is not correct to say here about symptoms development as this is just a consequence of the infection
- I highly recommend the authors to make a list of acronyms
- L 172 – better (insertions/deletions)
- L 215-216 – the sentence is not clear because of the fragments “these molecules” and “it accumulates”.
- L 228 – better “utilized for nucleic acid delivery”
- Figure 3
- should be moved upward and inserted closer to the first mentioning, for example before the last paragraph of section 4.2 or at least before the section 4.3.
- lacks the legend for c) and d) panels;
- moreover, I recommend designating vectors at panel b) differently, for example with the Roman numerals, it would make the legend to be more clear
- all acronyms used in the Figure should be explained in the legend
- Figure 4 is rather confusing. The name of the Figure if rather general, however, the Figure depicts the particular approach – CRISPR/Cas genome editing using viral vectors. It should be clearly reflected both in the Figure and in the legend. What do the authors mean as a gene of interest? How insect delivery is connected with Agrobacterial delivery? Systemic infection could be obtained not only using agrobacterial transfer, virions (including modified “transgenic” ones) are commonly used as well. Also it is not clear where panel a) ends and panel b) starts.
- L 461-462, 468, 567 – it is incorrect to use “protein” here: “Geminiviruses can easily integrate heterologous protein in their circular genome”; “larger proteins in geminivirus-based vectors”; “deliver heterologous proteins to plants”. Genes or other nt sequences are inserted in viral vectors and could be delivered into plant cell but not proteins. Should be corrected here and throughout the manuscript.
- L 497: “The genomes of positive-strand RNA viruses are always wrapped with nucleocapsid proteins, whereas negative-strand RNA genomes are not” What do the authors mean? The genome of - RNA viruses is also packed into virions.
- L 557 – “stable Cas9 lines” – should be corrected
- the supplementary table contains the information that could be included in the main text, thus, I recommend transferring it from the supplementary material to the body of the MS.
Reviewer 3 Report
Comments and Suggestions for Authors
This paper describes the use of geminivirus vectors for genome editing. This timely review covers the topic well and can be published in Viruses. However, I have some concerns that should be addressed before the paper can be accepted for publication.
Major concerns
- Section 5 “RNA viruses-derived vectors for genome engineering” is outside the scope of this review. The brief list of RNA viruses used for genome editing adds nothing to the main topic of the review, which is DNA-containing geminiviruses. Obviously, the intention of the authors was to provide information on RNA virus vectors for genome editing before comparing them to geminivirus-based vectors, as is done in Section 6. However, this comparison, at least as presented in Section 6, does not require a preliminary description of RNA virus vectors. Therefore, Section 5 should be removed from the paper, while Section 6 should be rewritten (see next comment).
- Section 6 has several serious problems.
- It is organized as a numbered list. Such a format should generally be avoided in scientific papers, so it should be converted to normal text.
- Not all of the seven points relate to advantages of geminiviruses as vectors. In fact, most of the considerations in this section are not related to these advantages. In many cases, the intended meaning is lost, making it difficult to interpret the information provided and to relate it specifically to advantages of geminiviruses as vectors.
- If the authors want to show the advantages of geminivirus vectors, they should also mention their disadvantages.
Therefore, I suggest changing the name of Section 6 to "Comparison of geminivirus-based vector for genome editing with other viral vectors" and CLEARLY present the advantages of geminivirus vectors in this section. I suggest using the form "First, .... Second,... etc". Next, it would be useful to list the disadvantages.
- Section 7 should definitely be deleted as irrelevant to scientific paper.
- The Abstract should be rewritten to bring it into good shape. My suggestions are as follows.
Lines 15-16. The first sentence – Do the authors mean plant viruses of viruses in general? “viral vectors can result … in experiments” – what does it mean? This sentence should be replaced with another one that has a clear meaning.
Lines 17-19. General statements about viruses taken from school textbooks should be deleted.
Line 20. “one of the largest” – is it? can be deleted
Line 24. “a world of” should be deleted
Lines 23-25. After this sentence (In this review, …), a more detailed description of the information provided in this review should be given.
Lines 26-28. (Many promoters …) This sentence is inappropriate for an abstract and should be deleted.
Line 28. “All” should be deleted.
- Lines 497-498. The genomes of positive-strand RNA viruses are always wrapped with nucleocapsid proteins, whereas negative-strand RNA genomes are not.
Incorrect statement, it is exactly the opposite of what it is. It appears that the considerations that follow this statement are also incorrect, although the exact meaning of these sentences is difficult to follow.
- Figure 1. Demonstration of geminivirus symptoms is not required in this paper. Figure 1A may be deleted.
- Figure 3 is overloaded. I suggest splitting it into two figures.
Author Response
We are thankful to the reviewer for their insightful comments and suggestions.
Comment 1. Section 5 “RNA viruses-derived vectors for genome engineering” is outside the scope of this review. The brief list of RNA viruses used for genome editing adds nothing to the main topic of the review, which is DNA-containing geminiviruses. Obviously, the intention of the authors was to provide information on RNA virus vectors for genome editing before comparing them to geminivirus-based vectors, as is done in Section 6. However, this comparison, at least as presented in Section 6, does not require a preliminary description of RNA virus vectors. Therefore, Section 5 should be removed from the paper, while Section 6 should be rewritten (see next comment).
Response: Section 5 has been deleted.
Comment 2: Section 6 has several serious problems.
- It is organized as a numbered list. Such a format should generally be avoided in scientific papers, so it should be converted to normal text.
Response: Section 6 has been renamed and numbered list also removed.
Comment 3: Not all of the seven points relate to advantages of geminiviruses as vectors. In fact, most of the considerations in this section are not related to these advantages. In many cases, the intended meaning is lost, making it difficult to interpret the information provided and to relate it specifically to advantages of geminiviruses as vectors.
- If the authors want to show the advantages of geminivirus vectors, they should also mention their disadvantages.
Therefore, I suggest changing the name of Section 6 to "Comparison of geminivirus-based vector for genome editing with other viral vectors" and CLEARLY present the advantages of geminivirus vectors in this section. I suggest using the form "First, .... Second,... etc". Next, it would be useful to list the disadvantages.
Response: section 6 named has been changed as suggested.
Comment 4: Section 7 should definitely be deleted as irrelevant to scientific paper.
Response: Figure 7 has been deleted as suggested.
Comment 5: The Abstract should be rewritten to bring it into good shape. My suggestions are as follows.
Lines 15-16. The first sentence – Do the authors mean plant viruses of viruses in general? “viral vectors can result … in experiments” – what does it mean? This sentence should be replaced with another one that has a clear meaning.
Response: The statement has been removed.
Lines 17-19. General statements about viruses taken from school textbooks should be deleted.
Response: The statement has been removed.
Line 20. “one of the largest” – is it? can be deleted
Response: ‘one of the largest’ has been deleted.
Line 24. “a world of” should be deleted
Response: ‘a world of’ has been deleted.
Lines 23-25. After this sentence (In this review, …), a more detailed description of the information provided in this review should be given.
Response: The detail has been added to the said sentence.
Lines 26-28. (Many promoters …) This sentence is inappropriate for an abstract and should be deleted.
Response: The statement has been deleted.
Line 28. “All” should be deleted.
Response: ‘All’ has been deleted.
Comment 6: Lines 497-498. The genomes of positive-strand RNA viruses are always wrapped with nucleocapsid proteins, whereas negative-strand RNA genomes are not. Incorrect statement, it is exactly the opposite of what it is. It appears that the considerations that follow this statement are also incorrect, although the exact meaning of these sentences is difficult to follow.
Response: This statement has been removed.
Comment 7: Figure 1. Demonstration of geminivirus symptoms is not required in this paper. Figure 1A may be deleted.
Response: Figure 1a has been deleted as suggested.
Comment 8: Figure 3 is overloaded. I suggest splitting it into two figures.
Response: Figure 3 has been divided into two as suggested.
Round 2
Reviewer 2 Report
Comments and Suggestions for Authors
The manuscript was significantly improved. There are only some minor issues left to be addressed:
- L 55 contains “to deliver heterologous proteins” which is incorrect, genes but not proteins are delivered (at least here)
- L 73: “to provide an understanding about” – seems to me as incorrect (English) and should be revised
- Figure 1, legend: all acronyms should be mentioned in the legend.
- L 179 – the sentence is interrupted with the figure – fix it, please
- Figure 2, legend – abbreviation “PAM” should be introduced; gRNA is not introduced anywhere in the text of the manuscript as well, thus should be inserted both in the legend and at the first mentioning
- L 217 – correct the spelling “polyethylene glycol (PEG)”
- L 216-217 – as to my knowledge, biolistic transformation is the same as particle bombardment; also PEG-mediated transformation is usually referred to protoplast transformation; “protoplast-mediated transformation” is also not correct – it should be “protoplast transformation” – thus the sentence should be thoroughly corrected
- Figure 3, legend contains “Rep protein” in context of replicon, that is not correct – replicon contains gene
- Figure 4 - the upper panel is of poor resolution and is hardly readable
- Table 1 – all species are in Latin except for tomato. Why? should also be Latin
- abstract and L 447 contain the phrase “ability of hijacking host genetic material” that seems not clear, probably better “ability to manipulate host genetic material”
- L 469 – acronym SNYV should be introduced
- Table 2 – all abbreviations for virus names should be introduced
- the Supplementary Table should have number S1 (now it is 2 or I have the old version); the reference to that table in the text should be “Table S1” instead of “supplementary file 1” – check throughout the MS.
